# Positive Psychology in Context of Peacekeeping Militaries: A Mediation Model of Work-Family Enrichment

**DOI:** 10.3390/ijerph18020429

**Published:** 2021-01-07

**Authors:** Carolina Silveira-Rodrigues, Maria José Chambel, Vânia Sofia Carvalho

**Affiliations:** CICPSI, Faculdade de Psicologia, Universidade de Lisboa, 2649-013 Lisbon, Portugal; carolinas2@campus.ul.pt

**Keywords:** peacekeeping mission, work-family enrichment, subjective well-being, health perceptions, satisfaction with life

## Abstract

Based on the work-family enrichment theory, this study analyzes the contribution of work-family and family-work enrichment to explain the military’s well-being during a peacekeeping mission. The data used were collected in a sample of 306 Brazilian soldiers, who were married and/or had children, during the phase named “employment of troops” (i.e., when peacekeepers had been in the Haitian territory and, as a result, away from their families, for between three to five months). Data analysis was performed using the Structural Equations Model. It was observed that the military’s perception of their spouses’ support for their participation during the mission had a positive relationship with both family-to-work enrichment and work-to-family enrichment, and the work-to-family enrichment mediated the relationship between the perception of the spouses’ support and the military’s health perception and general satisfaction with life. Theoretical and practical implications were discussed and limitations and suggestions for future research were presented.

## 1. Introduction

Since 1948, over one million men and women have been participating in peacekeeping operations under the aegis of the United Nations (UN). The military component in a UN peacekeeping operation seeks to create the necessary security conditions so that countries affected by serious conflicts can enjoy a permanent and lasting peace [1]. The United Nations Stabilization Mission in Haiti (MINUSTAH (Abbreviation derived from the franch language: Mission des Nations Unies pour la Stabilisation en Haïti.)), which began in 2004 and ended in 2017, was the most relevant peacekeeping mission in Brazil, considering both the duration of the mission and the number of military personnel involved. Approximately 35,500 servicemen were part of the Haiti Theater of Operations [2]. Even though participating in a UN peacekeeping mission is a voluntary decision, becoming a peacekeeper is an act that poses plenty of challenges, not only to the military personnel, but also to their whole families. The literature indicates that one of the most prevalent stressors in a peacekeeping operation is being away from family for a long time [3]. It is important to emphasize that the military profession always implies some withdrawal from family. However, peacekeeping missions have particular characteristics, such as the instability of the Theater of Operations environment and the uncertainty of the return date, which make distance from family even more relevant, not only for the personnel involved but actually for their entire family [4]. Hence, it is not surprising that research on the interface between work and family of militaries focused on work-family conflict [5,6,7].

However, a mission could also be a source of well-being for the militaries [8], and thus, analyzed from a positive perspective. For instance, a mission can be an opportunity for these soldiers to value their family relations more, as well as improve their individual and social skills, which can be useful for bettering the performance of their role in the family and at work [5]. In this vein, this study searches to take positive lenses to analyze the peacekeeping military mission experience. Peterson [9] defined positive psychology as the scientific study of what makes life most worth living, as such, our main goal was to contribute to understanding how a demanding experience to the individual (to be in a peacekeeping operation) may also have positive patterns. By doing so, we focused on the peacekeeping military’s work-family enrichment (WFE) [10], defined as the extent to which experiences in a role improve quality of life in another role [10]. This enrichment can occur bi-directionally and WFE manifests when resources gained at work improve the quality of life in the family role, though FWE occurs when resources gained in the family domain improve the quality of life in the work domain [10]. As such, our goal is to explore the WFE and FEW in the military context of a peacekeeping mission.

First, we explore a potential antecedent of WFE/FWE. It is well-established that supportive relationships are important in promoting life challenges [11,12,13] and social support has been demonstrated as an antecedent of WFE and FWE [14]. In particular, in a peacekeeping operation, the military is away from family this support can be crucial. However, as far as we know, there are few studies that show how family support, namely the spouse’s support, generates benefits that can be used at work, especially in the context of a military peacekeeping force. In this line, based on assumptions of Conservation of Resources Model (COR) [15,16,17] and on Work-Home Resources Model, developed by ten Brummelhuis and Bakker [18] that comprehend family support as an important resource to work and family dynamics. This study aims to investigate how the peacekeepers’ perception of their spouses’ support during the mission is understood as a resource and can related to their perception of WFE and FWE.

Second, we will also explore the relationship between WFE/FWE and the subjective well-being of these militaries in a peacekeeping mission. In accordance with the roots of positive psychology, which underlines strengths and health rather than weakness and illness, we explore as subjective well-being indicators the satisfaction with life and health perceptions [19]. Taken together, we aim to verify the role of WFE/FWE as a mediator between the perception of the spouses’ support and their subjective well-being.

In general, the purpose of this paper was to provide a more thorough picture of how a demanding work situation, that is, be in a peacekeeping force, may create positive paths to the military by summarizing the relationship between the two directions of work-family enrichment, their potential well-being consequences, and family support as a potential antecedent, as well as the mediator role of WFE and FEW. From the theoretical point of view, this study reinforces the importance that the positive work-family dynamics acquires in a specific organizational context. It contributes to the literature of positive psychology by demonstrating, through WFE, how positive synergies can be a relevant linking mechanism to promote subjective well-being and how support, through family, can generate these relationships.

From a practical point of view, the results obtained can enable the Brazilian Army (and Armed forces in general) to elaborate strategies to support the military families so that they understand the relevance of the mission and the importance of their support for the military well-being during the mission.

## 2. Family Support and Work-Family Enrichment

The WFE emphasizes that the work-family interface can generate gains to individuals, with “positive effect of experiences in one role on experiences or outcomes in another role’’ ([10], p. 74). Previous studies, theoretical and empirical, point out that enrichment “directions” should be examined independently; i.e., WFE occurs when resources gained at work improve the quality of family life, while FWE occurs when resources acquired in the family improve the quality of life at work [10]. The main idea is that both work and family domains provide individuals with resources such as higher self-esteem, salary, social support systems and other benefits that may help the individuals to perform better in other roles of their lives [20]. The transference of these resources can happen through two mechanisms: the instrumental and the affective. The first one produces direct influence, that is, the resources that are generated in one domain can directly influence the other domain. As for the affective mechanism, it is observed that the influences are indirect, that is, the experiences in a domain can arise a positive feeling in the form of enthusiasm or more energy that in turn will influence the behavior of the individual in the other domain.

The model of WFE [10] proposes the identification of five types of resources gained by the individual through work-family and family-work interactions, namely: Skills and perspectives (i.e., coping skills); psychological and physical resources (i.e., self-efficacy); socio-capital resources (i.e., networking); flexibility (i.e., work routines); material resources (i.e., money, bonuses). A peacekeeping mission can be a source of these resources that can benefit the military personnel in many aspects of their lives, including their family relationships. For example, Britt et al. [5] state that the peacekeeping mission in Lebanon was an opportunity for Norwegian soldiers to acquire skills such as stress tolerance, self-discipline, and self-control. These competencies, when transferred to the family, can help the soldiers in the resolution of conflicts and tensions that may occur in their family. In addition, Vietnam veterans described that the mission in this theater of operations enabled them to gain a range of psychological resources, such as improved self-esteem, assertiveness in social relationships, personal maturity, and the sense of responsibility when performing tasks [21]. These resources, when taken into the family sphere, can give the serviceman greater maturity to deal with challenges and difficulties that affect their family. When it comes to the acquisition of socio-capital resources in the mission, literature indicates a wide diversity of examples that refer to the expansion of the servicemen’s social network during a peacekeeping mission, allowing them to increase their professional contacts other than those established prior to the mission [21]. Moreover, the peacekeeper also has the opportunity to have unique cultural experiences and explore different social contexts [22]. This ability can be extended to the family by increasing a sense of general culture in the whole parental system, by transmitting knowledge from the peacekeepers to their dependents through storytelling and the presentation of images of the events that happened during the experience on the mission. With respect to flexibility features, Galantino and Ricotta [23] affirm that the constant changes inherent to military activities in the Theater of Operations, for instance, changes in the date of boarding flights, as well as other planning modifications that occur due to logistical demands, make the human resources employed in the mission need to adapt to complex and often unexpected situations. This important capacity acquired in this type of military operation can help individuals to better adapt to the demands of family roles, so that they can better balance parental or marital demands. Also, at the level of material resources, financial compensation is one of the benefits provided by the mission, which may bring monetary stability to the entire family system [24]. On the other hand, from a family-work perspective, family experiences may also contribute to the performance of operational activities on a peacekeeping mission. For instance, in humanitarian aid activities, in which the target audience is mainly composed of children and adolescents, parental experience can be a facilitator resource of these actions [25]. In fact, the family of origin has an important role in the transmission of personal values that can be applied in the professional context; namely the ability to share, the flexibility to solve tasks in general, manage relational conflicts and commitment, attributes that are appreciated in many contexts, not only personal but also professional [25]. From the perspective of a military peacekeeping force, these competencies acquired in the family can be extremely relevant, especially when analyzing the organizational environment of a theater of operations, due to the intense conviviality with other soldiers, which results from the confinement in the base and the moments of monotony of the work routine throughout the mission.

The contextual characteristics of each domain, namely, work support and family support, have been highlighted as antecedents of enrichment between work and family [26]. This support refers to the social support, that is, respect and protection, which an individual considers to have in each of these domains [27] and may have an emotional or instrumental character [28]. Emotional support comprises the behaviors or attitudes of family/work members that reflect the interest of the family/organizational system in various aspects of the individual’s professional/family life. Instrumental support refers to the behaviors and attitudes of the members of each domain, related to the management of tasks, in order to facilitate the individual’s life in the other domain. Family support refers to the support that family members give the individuals so that they can meet the demands of the job [26]. In the context of a peacekeeping mission, we consider the following as family support actions; the availability of a member of the family system to listen, discuss and advise the peacekeepers about their decision to participate in the mission or their life during the mission. The Work-Home Resources Model [18], to which the assumptions of Conservation of Resources Theory are applied [15,16,17] to the analysis of the work-family relationship, underpins the relevance of family support to promote WFE and FWE. COR is a motivational theory that explains human behavior based on the individuals’ evolutionary need to acquire and conserve resources for their survival [17]. It introduces the principle that individuals strive to obtain, retain, and protect everything they value on a personal, social and cultural level; i.e., material resources (financial resources), personal characteristics (self-esteem), or energy (physical disposition). Resources are important because they have value in themselves, but also because they serve to generate and obtain other resources that individuals’ value by becoming what Hobfoll [16], termed “the gain spiral”. In this regard, family support can be considered a resource of the family domain that causes the individuals to increase their personal resources to perform their professional role, thus generating WFE [18]. For example, the spouses’ emotional support can help the peacekeepers develop a positive sense of humor, as well as improve their self-esteem. These resources, when transferred to the work domain, can make the peacekeeper have a more vigorous and resilient attitude towards work [10]. In addition, this emotional support received from the spouse can also lead the soldier to feel fulfilled in the family domain, and as a result, can develop skills in this area, thus, generating FWE. For instance, the spouse’s emotional support can lead the soldier to develop active listening skills (instrumental channel) or a positive sense of affection (affective channel) that will facilitate the performance of their family role. In the meta-analysis study, developed by Lapierre and colleagues [14], which included 171 independent studies published between 1990 and 2016, authors found a positive and significant relationship between family support and WFE, as well as a positive and significant relationship between family support and FWE.

The present study considered family support represented by the perception that the peacekeepers have regarding their spouses’ support, as this member of the family is the main support for the soldier during a peacekeeping mission [4]. Therefore, in order to investigate family support for the WFE/FWE process, specifically in the context of military peacekeeping forces, we consider the following hypotheses:

**Hypotheses** **1** **(H1).**
*The peacekeepers’ perception of the spouses’ support for their participation during the mission had a positive relationship with FWE.*


**Hypotheses** **2** **(H2).**
*The peacekeepers’ perception of the spouses’ support for their participation during the mission had a positive relationship with WFE.*


## 3. Family Support and Well-Being: The Mediating Role of WFE and FWE

Within the body of literature on Positive Psychology studies, focused on subjective well-being, that is, a multi-faceted construct with affective and cognitive components [29], satisfaction with life and health perceptions are considered constructs included in subjective well-being [29]. In this line, the evaluation of overall satisfaction with life is a measure of subjective well-being recommended by the WHO [29] and refers to a cognitive process by which people broadly assess the quality of their lives in various domains, e.g., family, work. Individuals who positively assess overall satisfaction with life are satisfied with their life as a whole [30]. The health perceptions, as analyzed in this research, correspond to an overall assessment, not focused on specific health components (i.e., mental health, physical health, and physiological health). Hence, the health perceptions refer to the individuals’ explicit evaluation of their health, providing comprehensive information about the general state in which they evaluate their health, based on objective information, as well as how they feel and evaluate this information [19].

Given the aforementioned theory, namely Hobfoll’s COR Theory [15,16,17], individuals seek to maintain and protect resources and this maintenance and accumulation generate wellness. Further, people with resources are less likely to be affected by stressful circumstances that may negatively influence their well-being. When faced with stress, individuals with greater resources are more able to solve problems more effectively, and thus, are less likely to be affected by the depletion of resources that may occur during such situations [17].

Social support is a resource that promotes well-being in individuals, either because it promotes a sense of meaning and purpose in life or because it reduces the impact of stressors [31]. Holt-Lunstad and colleagues [32], through a meta-analysis study that included 148 studies with a total of 308,849 participants, showed that individual experiences in social relations significantly predicted a decrease in mortality risk, stressing the importance of social support for the individuals’ health and well-being. Likewise, social support in the work context has been considered a fundamental resource for adapting people to the demands of work, contributing to reduce the impact of occupational stressors and to increase workers’ well-being [33]. In fact, diverse studies have emphasized the importance of the perception of organizational support to explain workers’ well-being [34].

However, in addition, in the work context, the importance of family support to ensure the individuals’ well-being has been highlighted [32] family-support, or more specifically the spouse’s support, contributes to greater psychological well-being, as it helps to meet the needs of esteem, affiliation and emotional support of individuals [35]. Married individuals who have supportive spouses report greater well-being, feeling happier and more satisfied with their lives than individuals who are not married [36]. In the specific case of the military context, family social support attenuate the risk of posttraumatic stress disorder [37] and promote the military personnel’s adaptation to civilian life after the end of operations [38]. It should also be noted that the spouse’s support is essential to promote wellness, raising the peacekeepers’ resilience during a mission involving their removal from the family [39]. For example, Gewirtz, Polusny, DeGarmo, Khaylis and Erbes [40] argued that keeping contact with the spouse during military operations made the soldiers more focused on their work. Ferrier-Auerbach, Erbes, Polusny, Rath, Sponheim [41] showed that the spouses’ support was fundamental to avoid posttraumatic stress and depressive symptoms in the Army National Guard Brigade Combat Team soldiers in the Iraq Theater of Operations.

Furthermore, WFE/FWE are factors that triggers well-being in individuals [14]. As we noted earlier, according to the COR [16], individuals with greater resources are more able to solve problems and are less likely to be affected by the depletion of resources that may occur during stressful situations. As a result, in a situation of enrichment between work and family, individuals who have a “solid reservoir of resources” are better equipped to cope with stress and consequently feel more well-being [42]. In fact, individuals experiencing enrichment, either in the WFE direction or in the FWE direction, tend to report more satisfaction with work, more emotional commitment, more family satisfaction, more satisfaction with life [43], and more physical and psychological health [44]. As previously mentioned, the individuals’ perception of how much their families support them can promote innumerable resources that facilitate both family life and work life [45]. The emotional support received from the family will have positive effects on work-family enrichment and family-work enrichment, which in turn, have positive effects on the workers’ well-being [45]. In the present research, we hypothesize that, on the basis of accumulation of resources, the military personnel will have more FWE and WFE as a result of feeling supported by the spouse during their participation in the mission and in turn this perception of FWE and WFE, recognized as processes of obtaining more resources, will bring about greater general well-being. Therefore, we put forward the following hypotheses:

**Hypotheses** **3** **(H3).**
*WFE mediates the relationship between the perception of the spouse’s support for their participation during the mission and the peacekeeper’s individual well-being (i.e., satisfaction with life and health perception).*


**Hypotheses** **4** **(H4).**
*FWE mediates the relationship between the perception of the spouse’s support for their participation during the mission and the peacekeeper’s individual well-being (i.e., satisfaction with life and health perception).*


## 4. Method

### 4.1. Procedure and Participants

The sample of the present study was composed of Brazilian Army soldiers who integrated the United Nations peacekeeping mission in the 25th and 26th Army Infantry Battalions in the Peacekeeping Mission in Haiti from 10 December 2016 to 2 June 2017 and from 2 June 2017 to 15 October 2017, respectively. The data were collected through an online survey, and we obtained a total sample of 306, who were either married or had a civil union. The data were collected while the soldiers were in Haiti, i.e., during the phase called “employment”. When these peacekeepers answered the questionnaire, they had already been apart from their families for three to four months. All the respondents completed the survey anonymously and were assured that their answers would remain confidential by the researcher. There was no incentive (cash or the like) for participating in this study. The sample’s characteristics were: men = 293 (95.80%), woman = 13 (4.2%); less than 25 years old = 45 (14.7%); between 25 and 35 years old = 89 (29.1%); between 36 and 45 years old = 117 (31.2%); more than 45 years old = 55 (18%); without children = 84 (27.5%); with children = 222 (72.5%); between one and five years of tenure in army = 43 (14.1%); between five and ten years of tenure in army = 59 (19.3%); more than ten years of tenure in army = 204 (66.7%); without experience in a previous peacekeeping mission = 195 (63.7%); with previous experience in a previous peacekeeping mission = 111 (36.3%); attended middle school = 6 (2%); attended high school = 122 (39.8%) attended higher education = 88 (58.2%).

### 4.2. Measures

*Family Support.* To assess family support, we used four items from the *Family Support Inventory* for workers originally developed by King et al. [46]. The authors demonstrated the convergent discriminant, and nomological validity of the scale. Besides, the via viability of the instrument relative to other existing measures of social support and it was previously used with a Portuguese peacekeepers sample [47]. A sample item is “*Today, I see that my husband/wife is really supporting me in my mission here at BRABAT (Brazilian Battalion)*”. Items were scored on a 5-point rating scale from (1) totally disagree to (5) totally agree (*α* = 0.86).

*Family-Work Enrichment.* We measured FWE using the 9-item scale by Carlson and colleagues [48]. A sample item is “*My involvement with my family helps me to gain competencies and this helps me be a better soldier*”. Items were scored on a 5-point rating scale from (1) totally disagree to (5) totally agree (*α* = 0.92).

*Work-Family Enrichment*. We measured WFE using the 9-item scale by Carlson and colleagues [48]. A sample item is “My involvement with my work helps me to understand different points of view and this helps me be a better family member”. Items were scored on a 5-point rating scale from; (1) totally disagree to (5) totally agree (α = 0.96). It is important to note that the scale developed by Carlson and colleagues was validated with five samples by testing the content adequacy, dimensionality, reliability, factor structure invariance, convergent validity, divergent validity, and its relationship to work and family correlates [47]. Further, it was validated to Brazilian population and demonstrated adequate validity evidences [49].

*Well-being*. We measured peacekeepers’ general well-being with the assessment of health perceptions and satisfaction with life. The Health Perceptions Questionnaire developed by Ware and colleagues [19] was used to assess health perceptions. This scale demonstrated good internal-consistency and test-retest; reproducibility methods confirm the measurement reliability. Additionally, authors [19] highlighted that construct validation relied on theory and also did the empirical evidence about relationships among measures. The scale was composed of four-items, e.g., “*I am as healthy as others*”. Items were scored on a five-point rating scale from; (1) “definitively false” to (5) “definitively true” (*α* = 0.85). Satisfaction with life was assessed with the five-item scale by Diener et al. [30], which had already been used in Portugal [50], and also demonstrate psychometric properties to diverse Brazilian samples [51]. A sample of items is “*I am satisfied with my life*”. Items were scored on a seven-point rating scale from (1) “totally disagree” to (7) “totally agree” (*α* = 0.83).

*Control variables.* The variables having children (0 = No; 1 = Yes); previous participation in a peacekeeping mission (0 = No; 1 = Yes) and tenure in the Army (1 = between 1–5 years; 2 = between 5–10 years; 3 = More than 10 years) were introduced in the model as observable variables.

## 5. Results

### 5.1. Measurement Models and Descriptive Analysis

A test of the measurement model was conducted to control for common method variance and to establish discriminative validity. The one-factor model exhibited a poor fit to the data [χ^2^ (350) = 3435.17, *p* < 0.01, SRMR = 0.15, CFI = 0.52, IFI = 0.52, RMSEA = 0.17]. However, the five-factor model obtained an acceptable fit to the data [χ^2^ (340) = 755.99, *p* < 0.01, SRMR = 0.05, CFI = 0.94, IFI = 0.94, RMSEA = 0.06], significantly better than the one-factor model tested [Δχ^2^ (10) = 2679.18, *p* < 0.01], and all standardized regression coefficients were significant at the 0.001 level. These analyses revealed that the factor structures of the research variables were consistent with the conceptual model and that the manifest variables loaded, as intended, on the latent variables.

Table 1 presents the means, standard deviations, and correlation matrix obtained for the sample. The results found showed that servicemen had a positive perception of the support received from their spouse (*M* = 4.49; *SD* = 0.66; considering a Likert scale of five points). In addition, on average, peacekeepers perceive gains in the family domain that are being transferred to the work domain (FWE, *M* = 4.27; *SD* = 0.61; considering a Likert scale of five points), and vice-versa; i.e., on average, the soldiers perceived that peacekeeping missions contribute positively to their feelings of self-accomplishment, and this gain is transferred to the family domain (WFE, *M* = 4.00; *SD* = 0.68; considering a Likert scale of five points). The mean value registered for health perceptions indicates that servicemen had a positive perception of their health (*M* = 4.17; *SD* = 0.60; considering a Likert scale of five points), and a slightly positive level of satisfaction with their lives (*M* = 5.45; *SD* = 0.96; considering a Likert scale of seven points). With respect to the correlation matrix, the correlations are generally consistent with the theorized pattern of relationships.

### 5.2. Structural Models

To test our hypothesis, we started by testing a model with indirect effects through FWE and WFE and with no direct paths between family support and the well-being indicators, health perception and satisfaction with life (i.e., fully-mediated model). This model presented a good fit to the data [χ2 (343) = 780.64, *p* < 0.01, SRMR = 0.08, CFI = 0.93, IFI = 0.93, RMSEA = 0.07]. We further tested an alternative model with direct paths from family support to the two dimensions of well-being and we found that the partially-mediated model also provided a good fit to the data [χ2 (341) = 765.50, *p* < 0.01, SRMR = 0.06, CFI = 0.93, IFI = 0.93, RMSEA = 0.06], significantly better than the fully-mediated model [Δχ^2^ (2) = 15.14, *p* < 0.01]. Therefore, the partially mediated model was the one that best suited the data and was then chosen to test our hypothesis. To control for potential confounding effects, having children, previous participation in a peacekeeping mission and tenure in the Army were introduced in models as observed variables.

The results obtained with the partially mediated model showed (cf. Figure 1), as expected, that family support relates positively with both FWE (*β* = 0.44, *p* < 0.01) and WFE (*β* = 0.23, *p* < 0.01). Thus, hypotheses 1 and 2 were supported by the data.

Concerning the relationship between FWE and the two dimensions of well-being, the expected positive relationships were not significant (with health perception—*β* = 0.05, *n.s.*—non-significant; satisfaction with life—*β* = 0.10, *n.s.*). However, as predicted, the relationships between WFE and both health perceptions (*β* = 0.15, *p* < 0.05) and satisfaction with life (*β* = 0.25, *p* < 0.01) were positive and significant.

Regarding the role of FWE as a mediator, since the relationships between FWE and the two dimensions of well-being were found as not being significant, one of the conditions to test the mediation hypothesis was not verified (i.e., “*variations in the mediator significantly account for variations in the dependent variable”* ([52], p. 1176). With regard to the role of WFE as a mediator, this variable seems to be a partial mediator of the relationship between family support and health perceptions (*Z* = 1.71; *p* < 0.05), since the direct relationship between family support and health perceptions was significant (*β* = 0.19, *p* < 0.01). In addition, WFE seems to be a variable that contributes to totally explain the relationship between family support and satisfaction with life (*Z* = 2.38; *p* < 0.05). Given these results, our hypothesis 3 was partially supported by the data.

## 6. Discussion

The main purpose of the present research was to take positive lenses in analyzing a specific work situation in military life: To be in a peacekeeping mission. Specifically, the study tested the role of work-family enrichment as a mediator, posing the perception of the spouse’s support as potential antecedent and subjective well-being (i.e., satisfaction with life and health perceptions) as outcome. Our findings suggest that WFE has a mediating role in the relationship between spousal support and subjective well-being, since the perception of the spouse’s support may activate the perception that work enriches family, which is in turn, positively associated with health perception and satisfaction with life. Thus, this research study makes a significant contribution to the understanding of military well-being during a peacekeeping mission, as it suggests a model by specifying that a personal resource (e.g., spousal support) may have an indirect effect through WFE.

According to the COR theory’s assumption [15,16,17], namely the existence of resources that generate new resources that can be applied in another sphere of the individual’s life, we observed the existence of a significant relationship between the spouse’s support and the enrichment between work and family. In fact, reinforcing the idea that the spouse’s support is relevant in a situation of troop employment on a peacekeeping mission [4], we noticed that this resource is crucial for peacekeepers to consider that the relationship between their work and family lives ends up being reinforced because the experiences in one role improve the performance in the other [10]. In line with previous studies [14], we found a positive and significant relationship between spousal support and few. Highlighting that, when the soldiers consider that their spouses support them in their participation during the peacekeeping mission, they consider that their performance as family members is strengthened despite their departure, which contributes to their better performance in the professional field. Moreover, also in accordance with previous studies [14,43], we noted that the perception of being supported by their spouses contributed to the performance of their operational tasks, which in turn, would also favor the performance of their family role, even though they were distant from their relatives during troop employment. Also, in line with COR [17], we saw that the peacekeepers who considered themselves to have more resources (e.g., more WFE) were the ones who felt more subjective well-being. Actually, in accordance with previous studies [43,44,45], we also observed in this study that the fact that soldiers consider that their professional experience during the peacekeeping mission is contributing to a better performance of their family role makes them feel more satisfied with life and have a better perception of their health. More interestingly, according to the gain spiral principle postulated by COR [17], showing how the successive accumulation of resources—the spouse’s support and WFE—can contribute to improving individuals’ well-being. We perceived that the peacekeepers’ perception that their professional experience helps improve their family role during the mission is the mechanism that explains the relationship between spousal support and their well-being. In other words, when servicemen feel that their spouses support their participation in the peacekeeping mission, they experience more satisfaction with life and health, as this spousal support makes them feel that their professional life enriches the performance of their family role. For instance, a soldier feels that his/her spouse is available and shows a genuine interest in listening to him/her talking about the daily routine in the army. This spouse’s behavior may generate a feeling of being supported in the soldier, which puts him/her in a good mood and consecutively results in having a better performance both during the mission and as a family member. This feeling of having a better performance as a soldier and as a family member will contribute to higher levels of general well-being. In fact, previous studies have already suggested that WFE has a mediating role. For example, Nicklin and colleagues [53] found that WFE mediates the relationship between a supervisor’s support and general well-being. In the same vein, other researchers [54] also observed that WFE has a mediating role by explaining the relationship between perceived organizational support and general well-being.

Unlike what was expected and also what had been observed in previous studies [53], enrichment between family and work is not related to military well-being during the peacekeeping mission. This result may have occurred because the peacekeepers are estranged from family during the mission and, consequently, the professional dimension is much more preponderant as they are fully involved with the operational and administrative aspects of the mission. On the other hand, we found differences concerning the way spousal support is related to the dimensions of subjective well-being. More precisely, spousal support was found as having a direct relationship with the workers’ health perceptions, being the relationship between the two variables partially mediated by WFE. Since the relationship between spousal support and health perceptions was found as being only partially mediated by WFE, there may be other mediating variables that contribute to explain this relationship. In fact, previous studies have already shown the relationship between spousal support and health perceptions as being mediated by work-family conflict [55] and family-to-work conflict [56]. Overall, in line with the results obtained in the present study and on the basis of previous literature, future studies should include other mediating variables together with WFE in the same model. By doing so, it will be possible to observe to which extent all of these mediating variables jointly contribute to explain the relationship between spousal support and health perceptions.

Regarding the relationship between spousal support and life satisfaction, they seem to be only indirectly related due to WFE. As such, spousal support leads to higher WFE and, successively, higher WFE contributes to higher levels of life satisfaction among individuals. In other words, WFE seems to act as a mechanism that contributes to totally explain the relationship between spousal support and life satisfaction. Since the results are different to each subjective well-being component, this study reinforces the importance of understanding well-being through different dimensions [29].

### 6.1. Limitations and Future Studies

Our study has some limitations. Firstly, the data were transversal, collected in two battalions on a peacekeeping mission in the troop employment phase, which were in the theater of operations at different times. Therefore, the occurrence of causal interferences in relation to the variables cannot be inferred. Secondly, this research relies on self-report measures raising common method bias concerns. However, to minimize this effect, we followed several methodological and statistical recommendations made by Podsakoff and colleagues [57]. However, it should be reiterated that self-reported data seem to be the most appropriate way to capture the peacekeepers’ perceptions and assessment of these variables [58], and according to Spector [59], the concerns associated with the dependency of self-reported data may be overestimated. Another limitation is related to the sample, since all the participants were soldiers on a peacekeeping mission in the same theater of operations. It is, thus, necessary to be careful when generalizing the results in other contexts [60], namely missions of Brazilian Army soldiers’ missions in other countries, with different operational scenarios and different stressing factors. Finally, one limitation was the lack of record some information related with family life and to professional life that can influence the peacekeeper perception of family support, the relationship between work and family and his/her well-being. Therefore, we recommended that future studies should measure the frequency of contact with their family, some characteristics about the conjugal relationship (e.g., length of the relationship, whether the partner also works) and some working conditions, such as military group relationship’s, quality and task orientation, workload, perception of social recognition.

Despite the limitations, the results of this study underline several points for the improvement of theory, practice and research in the field of positive psychology through WFE/FWE and especially in military psychology applied to the context of peacekeeping missions. In the first place, by taking the positive side of a peacekeeping mission and by applying the WFE and FWE model [10], this study reinforces the importance of analyzing not only the negative facet of an event (i.e., to be in a peacekeeping mission), but also, the importance to envision the gains the same event.

By using the positive side of work-family relationship we provide support for the relationship between family support and subjective well-being and showed that the soldiers’ perception of how much their spouses support them is paramount for both WFE and FWE. In addition, the results highlight the importance of including the bi-directionality of WFE in future studies because its behavior to explain the relationship between family support and general well-being was distinct for WFE and FWE. Likewise, the results were different for the well-being variables, which reinforces the need for future studies to include the support given not only by the family but also by both: The supervisor and the peers. In fact, considering the importance of the influence of different kinds of social support in a peacekeeping mission and given that the soldier in such a situation is much more subject to organizational than to family members’ influences, the support from supervisor and colleagues are encouraged to future studies.

### 6.2. Practical Recommendations

From this study, we can take some practical recommendations for future peacekeeping missions. This study reinforces how resources, particularly spousal support, are important to keep the military subjective well-being and how the family dimension is salient in this process. Thus, it is of major relevance investments in family support guidelines by the Brazilian Army Institution. Since the preparation phase, it is of major relevance to evolving not only the military but also the militaries family. For example, recognizing mission as a hazard military’ family may be trained to identify coping mechanisms that they can use to deal with the withdrawal. Further, this study highlights that taking the positive side is also important, so families should also be informed how to understand this experience as a positive event and to help the military identifying the positive salience of events during the mission. For instance, if the military is expressing a negative emotion, they may support them to take a view the positive perspective of the situation. Overall, this study demonstrates that it is necessary to focus on positive practices that can increase spousal support during the peacekeeping mission [61].

## 7. Section

This study highlights that spousal support is crucial to peacekeeping militaries develop WFE. Moreover, WFE is an explain mechanism of the relationship between spousal support and subjective well-being. In general, it gives avenues to the well-being promotion of peacekeeping militaries.

## Figures and Tables

**Figure 1 ijerph-18-00429-f001:**
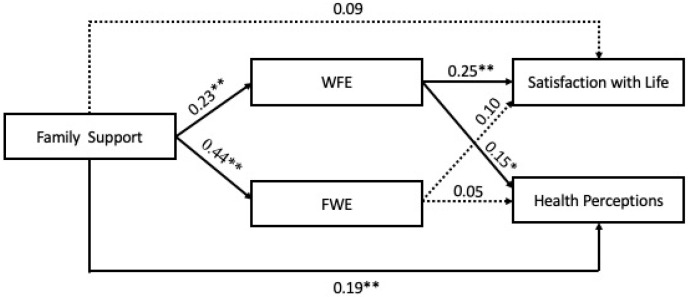
Standardized estimates for the partially-mediated model. ** *p* < 0.01; * *p* < 0.05.

**Table 1 ijerph-18-00429-t001:** Means, standard deviations, and correlations for the sample.

	*Mean*	*SD*	*r* Sample						
			1.	2.	3.	4.	5.	6.	7.
1. Have children	0.73	0.45							
2. Previous participation in peacekeeping mission	0.36	0.48	0.14 *						
3. Tenure in army	2.53	0.73	0.37 **	0.24 **					
4. Family support	4.49	0.66	0.04	−0.06	0.17 **				
5. FWE	4.27	0.61	0.02	−0.05	0.11 *	0.39 **			
6. WFE	4.00	0.68	−0.06	−0.06	0.02	0.21 **	0.58 **		
7. Health perceptions	4.17	0.60	−0.08	−0.02	−0.17 **	0.20 **	0.22 **	0.25 **	
8. Satisfaction with life	5.45	0.96	−0.01	0.05	0.11	0.16 **	0.26 **	0.31 **	0.26 **

Note. * *p* < 0.05; ** *p* < 0.01; FEW = Family-work enrichment; WFE = Work-family enrichment; Have children (0 = No; 1 = Yes); Previous participation in peacekeeping mission (0 = No; 1 = Yes); Tenure in army (1 = between 1–5 years old; 2 = between 5–10 years old; 3 = More than 10 years old).

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
