# Peer review of "Positive Psychology in Context of Peacekeeping Militaries: A Mediation Model of Work-Family Enrichment"

_ijerph, 2021, doi:10.3390/ijerph18020429_

Round 1
Reviewer 1 Report
IJERPH
Positive Psychology in Context of Peacekeeping Militaries: A Mediation Model of Work-Family Enrichment
The study is aimed at exploring the work-family enrichment (WFE) and family work enrichment (FEW) in the military context of a peacekeeping mission.
The topic is relevant and has applied implications.
Overall, this paper has many positive aspects (sound theoretical approach and methodological/statistical treatment) and can contribute to the body of literature about family/work enrichment.
Some limitations, that can be fairly easily addressed, should be addressed before publication.
pp.318-326: “we started by testing a model with indirect effects through FWE and WFE and with no direct paths between family support and the well-being indicators”. Were the well-being indicators considered independently or were they collapse into one unique indicator? It is unclear because there is no clear information.
Another unclear point is how/or if/ covariates were considered in the model.
The use of a figure of the final model would greatly facilitate the reading and the understanding of results.
Was frequency of contact with family measured? If yes, why not having included it as a covariate?
Author Response
The study is aimed at exploring the work-family enrichment (WFE) and family work enrichment (FEW) in the military context of a peacekeeping mission.
The topic is relevant and has applied implications.
Overall, this paper has many positive aspects (sound theoretical approach and methodological/statistical treatment) and can contribute to the body of literature about family/work enrichment.
Some limitations, that can be fairly easily addressed, should be addressed before publication.
Authors’ answer: We are grateful with this positive evaluation about our< manuscript.
pp.318-326: “We started by testing a model with indirect effects through FWE and WFE and with no direct paths between family support and the well-being indicators”. Were the well-being indicators considered independently or were they collapse into one unique indicator? It is unclear because there is no clear information.
Authors’ answer: In line with this suggestion, we included in the sentence that was considered two well-being indicators, namely health perceptions and satisfaction with life. We wrote “To test our hypothesis, we started by testing a model with indirect effects through FWE and WFE and with no direct paths between family support and the well-being indicators, health perception and satisfaction with life (i.e., fully-mediated model).”
Another unclear point is how/or if/ covariates were considered in the model.
Authors’ answer: We understand this comment and in line with the suggestion we added the sentence: “To control for potential confounding effects, having children, previous participation in a peacekeeping mission and tenure in the Army were introduced in models as observed variables.”
The use of a figure of the final model would greatly facilitate the reading and the understanding of results.
Authors’ answer: Accordingly, we add a figure with the results.
Was frequency of contact with family measured? If yes, why not having included it as a covariate?
Authors’ answer: In line with this suggestion (and others from reviewer 2), we included as another study limitation: “Finally, one limitation was the lack of record some information related with family life and to professional life that can influence the peacekeeper perception of family support, the relationship between work and family and his/her well-being. Thus, we recommended that future studies should measure the frequency of contact with their family, some characteristics about the conjugal relationship (e.g. length of the relationship, whether the partner also works) and some working conditions, such as military group relationship's, quality and task orientation, workload, perception of social recognition.”
Reviewer 2 Report
The research topic is really interesting and relevant, I hope that the following suggestions will help you improve your work
- Authors could expand the literature including more recent references
- authors should better describe sample's characteristic
- Did the authors consider some characteristics of the relationship (e.g. length of the relationship, whether the partner also works, frequency and length of separation periods, etc.)? Why did they choose not to include them among the factors analyzed as control variables?
- Authors should explain a little more the scales and the reasons for their choices
- About the statement from line 390 to line 400 I would suggest to authors to consider also other working dimensions such as military group relationship's quality and task orientation, workload, perception of social recognition...
- I would suggest to consider for future studies to consider also more "organizational" factors (see the comment above)
Author Response
The research topic is really interesting and relevant, I hope that the following suggestions will help you improve your work
- Authors could expand the literature including more recent references
Authors’ answer: We take the Reviewer suggestion into consideration. We did research to find recent literature and we add two references. However, we found it difficult to improve with more recent findings since our study is with a specific population – peacekeeping militaries – without more recent studies in these issues.
The new references included are:
Wayne, J. H.; Matthews, R.; Crawford, W.; Casper, W.J. Predictors and processes of satisfaction with work-family balance examining the role of personal, work and family resources and conflict and enrichment. Hum Resour Manag, 2020, 59, 25-42. DOI: https://doi.org/10.1002/hrm.21971
Mills, M., Tortez, L.M. Finting for family: Considerations of work-family conflict in military service member parents. In Occupational Stress and Well-Being in Military Contexts (Research in Occupational Stress and Well Being. Harms, P. D., Perrewé P. L.(Eds.). Emerald Publishing Limited. 2018, pp. 91-116.
- authors should better describe sample's characteristic
Authors’ answer: In line with this suggestion we wrote a new sample´s description: “The sample´s characteristics were: men = 293(95.80%), woman = 13 (4,2%); less than 25 years old = 45 (14,7%); between 25 and 35 years old = 89 (29,1%); between 36 and 45 years old = 117 (31,2%); more than 45 years old = 55 (18%); without children = 84 (27,5%); with children = 222 (72,5%); between one and five years of tenure in army = 43 (14,1%); between five and ten years of tenure in army = 59 (19,3%); more than ten years of tenure in army = 204 (66,7%); without experience in a previous peacekeeping mission = 195 (63,7%); with previous experience in a previous peacekeeping mission = 111 (36,3%); attended middle school = 6 (2%); attended high school = 122 (39,8%) attended higher education = 88 (58,2%).”
- Did the authors consider some characteristics of the relationship (e.g. length of the relationship, whether the partner also works, frequency and length of separation periods, etc.)? Why did they choose not to include them among the factors analyzed as control variables?
Authors’ answer: These characteristics were not measured but we considered this failure as a limitation of the study and wrote: “Finally, one limitation was the lack of record some information related with family life and to professional life that can influence the peacekeeper perception of family support, the relationship between work and family and his/her well-being. Thus, we recommended that future studies should measured the frequency of contact with their family, some characteristics about conjugal relationship (e.g. length of the relationship, whether the partner also works) and some working conditions, such as military group relationship's, quality and task orientation, workload, perception of social recognition.”
- Authors should explain a little more the scales and the reasons for their choices
Authors’ answer: Following the reviewer suggestion we added some more information about the scales that justified their use in this study:
In Family Support Scale we added: “The authors demonstrated the convergent discriminant, and nomological validity of the scale. Besides, the via viability of the instrument relative to other existing measures of social support”
In the Work-family enrichment scale we added: “It is important to note that the scale developed by Carlson and colleagues was validated with five samples by testing the content adequacy, dimensionality, reliability, factor structure invariance, convergent validity, divergent validity, and its relationship to work and family correlates [48]. Further, it was validated to Brazilian population and demonstrated adequate validity evidences [49].”
In General Health Perceptions Scale we added: This scale demonstrated good internal-consistency and test-retest; reproducibility methods confirm the measurement reliability. Additionally, authors [18] highlighted that construct validation relied on theory and also did the empirical evidence about relationships among measures.
In Satisfaction with life scale we added: “and also demonstrate psychometric properties to diverse Brazilian samples [51]”
- About the statement from line 390 to line 400 I would suggest to authors to consider also other working dimensions such as military group relationship's quality and task orientation, workload, perception of social recognition...
- I would suggest to consider for future studies to consider also more "organizational" factors (see the comment above).
Authors’ answer: In line with this comment and suggestion, as you can see in our previous answer, we recommended that future studies – in limitations and future studies section - will include some working conditions.
Round 2
Reviewer 1 Report
Authors made all required modifications.
However, although they did insert a figure, the figure was not in the text.